# Exploring people's thoughts about the causes of ethnic stereotypes

**Anita Schmalor, Benjamin Y. Cheung, Steven J. Heine** *

Department of Psychology, University if British Columbia, Vancouver, British Columbia, Canada

* heine@psych.ubc.ca

## Abstract

Much research has shown that people tend to view genes in rather deterministic ways—often termed genetic essentialism. We explored how people would view the causes of ethnic stereotypes in contexts where human genetic variability was either emphasized or downplayed. In two studies with over 1600 participants we found that people viewed ethnic stereotypes to be more of a function of underlying genetics after they read an article describing how ancestry can be estimated by geographic distributions of gene frequencies than after reading an article describing how relatively homogeneous the human genome was or after reading a control essay. Moreover, people were more likely to attribute ethnic stereotypes to genes when they scored higher on a measure of genetic essentialism or when they had less knowledge about genes. Our understanding of stereotypes is a function of our understanding of genetics.

## Introduction

Csanád Szegedi, then member of the European Parliament, converted to living his life as an Orthodox Jew at the age of 30. What makes his conversion remarkable is that shortly before his conversion he was the vice president of the anti-Semitic Jobbik Party in Hungary, and had published a book full of anti-Semitic sentiment. So why would such a person convert to Judaism? It appears to have been based on something that he learned about his own ancestry. Szegedi reports having been stunned to learn that his maternal grandmother was Jewish, a fact she had kept secret [1]. That is, Szegedi had learned that he, too, was Jewish by descent. Szegedi's dramatic transformation underscores the important role that our understanding of our biological roots play in shaping who we think we are. Despite being raised as a Christian, and not having any Jewish experiences, Szegedi came to identify himself in line with his genetic ancestry [2].

Szegedi is not alone in viewing his genetic ancestry as the key to his identity. Consumer genomic companies inform people about the possible geographic origins of their ancestors, and many people have reacted to their test results by changing how they identify themselves, such as choosing different ethnic identities when completing a census, joining new communities, cheering for different national sports teams, and learning new languages [see 3]. These rather dramatic reactions from the results of a genetic test of somewhat dubious accuracy [for critical reviews of genetic ancestry testing see 4–6] are telling. They suggest that people

**Data Availability Statement:** The data files, codebooks, and analysis scripts underlying this study are available on the OSF (https://osf.io/rdqw3/).

**Funding:** SJH 435-2019-0480 Social Sciences and Humanities Research Council https://www.sshrc-crsh.gc.ca.

**Competing interests:** No competing interests exist.

understand genes to be relevant to their ethnic identities, and it raises the question of whether they believe genes to be relevant to the ethnic identities of others, more generally.

People do tend to be interested in the ethnic identity of others [e.g., 7, 8], especially those who score higher on measures of prejudice [9], and they may identify tendencies that they believe covary with different ethnic groups [e.g., 10]. For example, a visitor to France may well notice the sophistication and taste of French cuisine, or a visitor to Japan may be struck by the orderliness of public behavior. But what do people believe is the cause of these ethnic differences? Do they think that they are largely the result of people's cultural experiences, or do they think of them as the result of innate, genetic factors? Moreover, how does learning about population variation in genetics affect people's views about the underlying causes of ethnic differences? In this paper, we explore how people's theories about the bases of ethnic differences are affected by encounters with scientific descriptions of population variation in gene frequencies.

## Psychological essentialism

Why did Szegedi convert to Judaism when he learned about his Jewish ancestry? Or, more generally, why do people often view genes as holding the key to their identity [3, 11]? A consideration of psychological essentialism may shed light on these questions. Psychological essentialism describes a set of intuitions that people have which leads them to think of the natural world to emerge as it does as the result of an underlying hidden essence [e.g., 12]. People have a variety of intuitions about essences: They are thought to be the ultimate cause for a specific outcome [e.g., 13], they are believed to be stable over time [e.g., 14], they are considered to be immutable, even if superficial characteristics are altered [15, 16], they are seen to be the bases of categories, underlying both different species of animals and different human groups [17], and they are believed to be primarily relevant for natural kinds, and not artifacts [15]. These intuitions guide the way that people make sense of the natural world.

## Genetic essentialism

An influential account of psychological essentialism proposes that people struggle to form concrete mental representations of what essences actually are, and they come to rely on essence placeholders upon which they project their intuitions about essences [18]. While people have relied upon various essence placeholders over time, such as viewing the four humors of Hippocrates to be the key to health and personality, an especially suitable essence placeholder is people's understandings of genes. Genes are often understood as ultimate causes, immutable, natural, and they carve up the social world into homogeneous and discrete categories, a tendency termed "genetic essentialism" [19]. Hence, when people come to learn that genes are involved in a particular trait, they come to view that trait in more essentialist ways [for a review see 20]. For example, when people read about the existence of "obesity genes" they come to view their weight as more beyond their control [21], or when people are informed that a violent criminal possesses a "warrior gene" they view him as less responsible for his crimes [22].

While there are indeed genetic causes that can operate in highly deterministic ways that resemble these essentialist tendencies, such as fully penetrant monogenic conditions (e.g., Huntington's disease), the so-called "fourth law of behavioral genetics" states that "a typical human behavioral trait is associated with very many genetic variants, each of which accounts for a very small percentage of the behavioral variability" [23, p. 305]. For example, to account for the genetic variability of both human height and IQ, two highly heritable traits, would each require hundreds of thousands of common genetic variants [24, 25]. Moreover, the expression of these many genes is guided by environmental experiences and is further moderated by various epigenetic markers across a developmental trajectory [see 4, for a review]. In sum, the

typical ways that genotypes influence phenotypes are vastly more complex than our simple essentialist intuitions would suggest.

## Genetic essentialism and perceptions of ethnicity

How might people's genetic essentialist biases affect how they understand ethnicity? While the field of cultural psychology has documented the many ways that people's cultural experiences shape their ways of thinking [e.g., 26, 27], it is not uncommon for people to assume that some distinctions between different ethnic groups are grounded in an underlying biological essence [28, 29]. An essentialist view of ethnic identity suggests that people's views of ethnicity may be influenced by information that calls attention to any underlying genetic differences between populations. If one reflects on the fact that people living on one continent are more likely to possess some genetic variants than people living on another continent, this may highlight how people living on those two continents belong to discrete categories, and have fundamental differences [14, 17]. Indeed, there is much research to suggest that people's understandings of ethnic identities are influenced by discussions regarding genes.

One study presented German students with either an essay that described how geographic ancestry could be revealed by genomic analyses or a control essay on an unrelated topic [30]. Participants were later asked questions about their views on expanding the European Union, and on their feelings towards people from various Western European and Eastern European countries. Those participants who read the essay about genes and ancestry showed a more marked ingroup preference for Western Europeans over Eastern Europeans compared with those who read the control essay. It seems that a consideration of the genetic foundation of ancestry led people to have more preferences for those who shared their own ancestry. Similarly, people who read arguments that the human population's genome varied significantly tended to evaluate ingroup and outgroup faces in a more dichotomous way compared with those who read that there was little genetic variation across the human species [31; also see 32]. In addition, a study of American Jewish participants compared people who read essays arguing that Jews and Arabs were either highly genetically similar or that they were genetically distinct. Those who read about the genetic similarities between Jews and Arabs indicated more support for peacemaking efforts in the Middle East than those who had read about the genetic differences between these groups [33]. The common finding across these studies is that discussions about genetic differences between populations is associated with tendencies to think of those populations as fundamentally distinct. Converging findings have also been observed by comparing people who read essays arguing that race is a biological construct in contrast to those who read a social account of race; those who are led to focus on a biological basis of race show more evidence of prejudice [34], and less of a tendency to think of Asian-Americans as Americans [35, 36]. These studies highlight how ethnic differences are viewed as more pronounced when biological factors are considered [also see 37].

While the extant literature has found that genetic information can affect the ways people think about different ethnicities, we explore whether thinking about genetic differences between different populations leads people to think of *any* differences between these populations as being more likely the result of genes. Because genes are often perceived as carving up the social world into homogenous and discrete categories [19], different populations may be seen as fundamentally different which might increase the psychological distance people feel towards them.

Specifically, how might people think about ethnic stereotypes when genes are brought into the discussion? People can readily describe different ethnic stereotypes, but it is unclear why people believe those stereotypes exist. Are they the product of people's different cultural

experiences, or are they grounded in the different genes that people possess? Because people's genetic essentialist biases make them think of social categories as more homogeneous and discrete [19], it would seem that a consideration of genes would make ethnic stereotypes appear to be more likely to have a genetic basis. For example, one study found that informing people that DNA tests can measure one's racial ancestry led people to have a more reified view of race [38]. To explore this question, we investigated across two studies how people attributed the cause of different ethnic stereotypes when they were presented with accurate scientific arguments highlighting either the homogeneity or the heterogeneity of gene frequencies from around the world.

## Study 1

In Study 1, we aimed to test whether participants who are led to believe that there is much genetic variation across different human groups would attribute various stereotypes about different ethnic groups more to genetic causes (and less to environmental causes) compared to people who are led to believe that different human groups show little genetic variation. We also included a control condition in which people learned about neither of these perspectives to test which of the two experimental conditions is closer to people's default thoughts about human genetic variation. We also explored whether participants who score higher on an individual difference measure of genetic essentialist tendencies would attribute various stereotypes about ethnic groups more to genetic and less to environmental causes. Last, because social dominance orientation is associated with more prejudicial attitudes towards outgroups [39], we also explored how this variable would relate to people's understandings about the causes of ethnic stereotypes.

### Method

**Participants.** Using G*Power [40], we calculated the minimum sample size needed to detect a significant effect with 0.80 power, assuming a small-to-medium effect size between three groups ($d$ = 0.30). This yielded a recommended sample size of 336. We collected data from 425 Americans through MTurk in case some participants do not pass comprehension checks (discussed later; $M$ age = 33.55, $SD$ = 10.75; 52% male, 47% female, 1% other; 67% Caucasian, 6% Black/African American, 6% Hispanic, 5% Asian, 4% mixed, 3% other). Participants provided informed written consent and the studies received ethical approval from the University of British Columbia Behavioral Research Ethics Board. All participants were over 18.

**Measures.** *Manipulation*. Participants were randomly assigned to read one of three articles, each of which was scientifically accurate. Those assigned to the Genetic Differences condition read an article that argued that people's geographic ancestry could be estimated through their genes. Those assigned to the Genetic Similarities condition read that the human genome is unusually homogenous, much less variable than that of chimpanzees, and that we all share common ancestors. In contrast, those in the Control condition read an article that described ways to improve home decoration (see SOM for articles). Following the article, participants were then asked 5 multiple choice questions about the content of the article as comprehension checks.

*Perceived accuracy of the ethnic stereotypes and attribution to environmental and genetic causes*. Participants saw a list of 10 stereotypes about different ethnic groups (e.g., "Japanese have longer lifespans than people from most other countries," "The Dutch are, on average, taller than people from most other countries;" see S1 Table in S1 File SOM for complete list). We generated this list of stereotypes from internet searches, discussions with various individuals, and by reading a list of racial stereotypes [41]. After reading each stereotype, participants were

asked to indicate on a scale from 0% to 100% to what extent each stereotype could be attributed to genetic causes ($M$ = 40.73%, $SD$ = 18.29%). Likewise, they were then asked on the same scale the extent to which each stereotype could be attributed to environmental causes ($M$ = 65.67%, $SD$ = 17.64%). Then participants had to indicate how accurate they found the stereotype to be on a 7-point scale from "completely inaccurate" to "completely accurate" ($M$ = 4.15, $SD$ = 0.91).

*Genetic essentialism*. Participants indicated to what extent they viewed genes as the essence of traits and behavior on the 24-item Genetic Essentialism Tendencies Scale [42] on a 5-point scale from "strongly disagree" to "strongly agree" ($M$ = 2.71, $SD$ = 0.60).

*Social Dominance Orientation (SDO)*. Participants indicated on a 16-item scale to what extent they viewed some groups as inferior to others on a 7-point scale from "strongly disagree" to "strongly agree" [$M$ = 2.23, $SD$ = 1.18; 39].

## Results and discussion

We first explored whether participants had completed the 5 comprehension checks correctly. Anyone who got more than 2 incorrect was excluded from analyses, which resulted in the exclusion of 7 participants. On average, participants answered 4.77 of the 5 questions correctly, indicating good comprehension. We further included two questions about their opinion on different aspects of the article which were intended to distract them from the true purpose of the manipulation and were not analyzed. Moreover, because it is meaningless to consider what the causes are of stereotypes if one does not believe the stereotype to be accurate, we only analyzed the data for stereotypes that a participant indicated they believed was accurate (i.e., the participant had to rate the accuracy of that particular stereotype at or above the midpoint of the scale). On average, participants perceived 69.59% ($SD$ = 25.32) of the stereotypes to be accurate. For 3 participants (0. 72%), none of the stereotypes reached the minimum accuracy score, and so they were not included in the analyses (see S1 Table in S1 File SOM for mean accuracy, mean genetic, and mean environmental attributions by stereotype). Note that there was no significant difference in believing the stereotypes to be accurate between conditions, $F$(2, 414) = 1.50, $p$ = .224, $\eta^2$ = .007. To compare the extent to which stereotypes are attributed to genetic and environmental causes between the three conditions, we conducted two ANOVAs and used Holm's corrections for multiple comparisons. The difference between conditions in attributing ethnic stereotypes to genetic causes was significant, $F$(2, 411) = 5.13, $p$ = .006, $\eta^2$ = .02. As hypothesized, participants in the Genetic Differences condition attributed the ethnic stereotypes more to genetic causes ($M$ = 47.58%, $SD$ = 20.13%) than did participants in the Genetic Similarities condition ($M$ = 40.73%, $SD$ = 18.89%), $p$ = .014, $d$ = 0.35. Participants in the Control condition also attributed ethnic stereotypes more to genetic causes ($M$ = 47.51%, $SD$ = 19.87%) than did participants in the Genetic Similarities condition, $p$ = .014, $d$ = 0.35. There were no significant differences between the Control and Genetic Differences condition, $p$ = .976, $d$ = 0.003 (Fig 1).

The difference between conditions in attributing ethnic stereotypes to environmental causes was also significant, $F$(2, 411) = 4.53, $p$ = .011, $\eta^2$ = .02. Participants in the Genetic Differences condition attributed the ethnic stereotypes marginally less to environmental causes ($M$ = 67.46%, $SD$ = 17.49%) than participants in the Genetic Similarities condition ($M$ = 71.72%, $SD$ = 14.85%), $p$ = .091, $d$ = 0.26. Participants in the Control condition attributed ethnic stereotypes less to environmental causes ($M$ = 65.38%, $SD$ = 18.96%) than did participants in the Genetic Similarities condition, $p$ = .009, $d$ = 0.37. There were no significant differences between the Control and Genetic Differences condition, $p$ = .306, $d$ = 0.11. The null effects between the Control and Genetic Differences condition suggests that the default

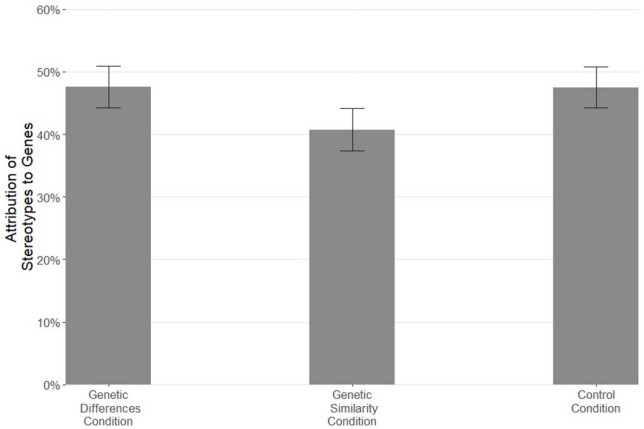

**Fig 1. Mean attribution of stereotypes to genes by condition.** Error bars are 95% confidence intervals.

position of people in the Control condition was to assume that different populations are somewhat genetically distinct.

We correlated people's attributions for the ethnic stereotypes with their scores on the genetic essentialism tendencies measure and the social dominance orientation measure. The higher participants scored on genetic essentialist tendencies, the more they attributed the ethnic stereotypes to genetic causes, $r = .41$, $p < .001$, and the less they attributed them to environmental causes, $r = -.16$, $p = .002$. Likewise, the higher participants scored on social dominance orientation, the more they attributed the ethnic stereotypes to genetic causes, $r = .20$, $p < .001$. On the other hand, there was no significant correlation between social dominance orientation and attributions to environmental causes, $r = -.004$, $p = .379$ (see Table 1: for these analyses, all participants were included in these analyses, even if they rated the accuracy of a stereotype to be below the midpoint of the scale).

In sum, the results provide initial support for our hypothesis. People who were led to reflect on the homogeneity of the human genome were less likely to think of ethnic differences as the product of underlying genetics. Participants in the control condition were similar to those in the Genetic Differences condition, suggesting that this may be consistent with people's default

**Table 1. Correlations between perceived accuracy of stereotypes, attribution of stereotypes to genes, environment, genetic essentialism, and social dominance orientation.**

|  | Perceived Accuracy of Stereotypes | Attribution of stereotypes to genetic causes | Attribution of stereotypes to environmental causes | Genetic Essentialism Tendencies | Social Dominance Orientation |
|---|---|---|---|---|---|
| Perceived Accuracy of Stereotypes |  | 0.36($< .001$) | 0.27($< .001$) | 0.29($< .001$) | 0.23($< .001$) |
| Attribution of stereotypes to genetic causes | 0.36($< .001$) |  | -0.28($< .001$) | 0.41($< .001$) | 0.20($< .001$) |
| Attribution of stereotypes to environmental causes | 0.27($< .001$) | -0.28($< .001$) |  | -0.16(.002) | -0.04(.379) |
| Genetic Essentialism Tendencies | 0.29($< .001$) | 0.41($< .001$) | -0.16(.002) |  | 0.29($< .001$) |
| Social Dominance Orientation | 0.23($< .001$) | 0.20($< .001$) | -0.04(.379) | 0.29($< .001$) |  |

Computed correlation used pearson-method with listwise-deletion.

theories about genetic variation. We also found that participants who believe in genetic essentialism attributed the ethnic stereotypes more to genetic and less to environmental causes, suggesting that people's understanding of the role of genes is associated with how they think about ethnic stereotypes.

## Study 2

In Study 2, we aimed to replicate our key finding that people who are led to believe that there is much genetic variation across the human species attribute stereotypes about ethnic groups more to genetic causes. The study was identical to Study 1, except we expanded the list to include 20 ethnic stereotypes. We also included an additional measure of genetic essentialist tendencies, and we explored the role of knowledge about genes, and the effect that the manipulation may have on feelings towards the different ethnic groups.

## Method

**Participants.** Since Study 1 results suggest a small but significant effect of the independent variable, we adjusted our sampling accordingly. Using G*Power [40], we calculated the minimum sample size needed to detect a significant effect with 0.80 power, assuming a small effect size between three groups ($d = 0.20$). This yielded a recommended sample size of 969. We collected data from 1238 Americans through MTurk in case some participants did not pass comprehension checks ($M$ age = 34.53, $SD = 11.16$; 54% female, 45% male, 1% other; 69% Caucasian, 12% mixed, 8% Asian, 7% Black/African American, 2% Native American, 2% other). Participants provided informed written consent and the studies received ethical approval from the University of British Columbia Behavioral Research Ethics Board. All participants were over 18.

**Measures.** *Manipulation*. Participants were randomly assigned to read one of the same three articles as in Study 1. We again followed this with 5 comprehension check items.

*Perceived accuracy of the ethnic stereotypes and attribution to environmental and genetic causes*. Participants saw a list of 20 stereotypes about different ethnic groups (e.g., "On average, French people have a more refined palate than people from most other countries, and they prefer the taste of high quality cuisine."; see S2 Table in S1 File in the SOM for complete list). The stereotypes touched upon a broad array of traits, and many of them were distinctly negative. Participants were asked directly after reading each stereotype how offensive they found it on a 7-point scale from "not offensive at all" to "completely offensive" ($M = 2.98$, $SD = 1.21$). We reasoned that if participants were given the chance to express their discomfort with these stereotypes, they might be more willing to consider the accuracy of them. Participants then were asked to indicate how accurate they found the stereotype to be on a 7-point scale from "completely inaccurate" to "completely accurate" ($M = 3.60$, $SD = 0.92$). Following this, participants were asked to indicate on a scale from 0% ("not at all") to 100% ("entirely") to what extent each stereotype (assuming it was true) can be attributed to genetic ($M = 34.82\%$, $SD = 19.04\%$) and to environmental causes ($M = 69.01\%$, $SD = 17.51\%$). Since we were not interested in how offensive participants found the stereotypes we did not analyze this question further.

*Genetic essentialism*. We assessed genetic essentialist beliefs in two ways. As in Study 1, participants responded to the Genetic Essentialist Tendencies Scale [$M = 2.71$, $SD = 0.58$; 42]. Participants also completed the Belief in Genetic Determinism Scale [30] on a 7-point scale from "strongly disagree" to "strongly agree" ($M = 4.10$, $SD = 0.88$; Note that we only have 16 items for the Belief in Genetic Determinism rather than 18 due to a copying and pasting error in the survey).

*Social Dominance Orientation (SDO)*. Participants indicated to what extent they viewed some groups as inferior to others on a 16-item scale from "strongly disagree" to "strongly agree" [$M$ = 2.43, $SD$ = 1.22; 39].

*Genetics knowledge.* Participants responded to 9 questions about different aspects of genes —some were created by us, while others were adapted from [43]. We added up the number of correct responses ($M$ = 6.28, $SD$ = 1.46).

*Attitudes towards different ethnic groups.* We asked participants how they felt towards the different ethnic groups about which they saw the stereotypes on a feeling thermometer from "very cold or unfavorable feeling" (0 degrees) to "very warm or favorable feeling" (100 degrees) ($M$ = 70.09, $SD$ = 16.89).

## Results and discussion

First, we analyzed how participants did on the 5 comprehension check items. On average participants scored 4.75 out of 5, indicating good comprehension. As in Study 1, we excluded anyone who got more than 2 incorrect, which resulted in the exclusion of 15 participants. As in Study 1, we only analyzed participants' attributions to genetic and environmental causes if they viewed the stereotype as accurate (i.e., they had to score at or above the midpoint on the scale). On average, participants perceived 54.02% ($SD$ = 24.35) of the stereotypes to be accurate. For 15 participants (1.23%), none of the stereotypes reached the minimum accuracy score, and so they were not included in the analyses (see S2 Table in S1 File SOM for mean accuracy, mean genetic, and mean environmental attributions by stereotype). Note that, as in Study 1, there was no significant difference in believing the stereotypes to be accurate between conditions, $F(2, 1173)$ = 1.63, $p$ = .196, $\eta^2$ = .003.

To compare the extent to which stereotypes are attributed to genetic and environmental causes between the three conditions, we conducted two ANOVAs and used Holm's corrections for multiple comparisons. The difference between conditions in attributing ethnic stereotypes to genetic causes was significant, $F(2, 1157)$ = 7.07, $p < .001$, $\eta^2$ = .01. Replicating Study 1, participants in the Genetic Differences condition attributed the ethnic stereotypes more to genetic causes ($M$ = 43.47%, $SD$ = 19.68%) than participants in the Genetic Similarities condition ($M$ = 38.63%, $SD$ = 21.00%), $p$ = .002, $d$ = 0.24. Unlike in Study 1, participants in the Control condition attributed ethnic stereotypes less to genetic causes ($M$ = 38.93%, $SD$ = 20.80%) than did participants in the Genetic Differences condition, $p$ = .005, $d$ = 0.22. There were no significant differences between the Control and Genetic Similarities conditions, $p$ = .841, $d$ = 0.01 (Fig 2). The similarities between the Control and Genetic Similarities condition suggests that people's default perspective in the Control condition was to be thinking along the lines that different populations are quite genetically similar.

The difference between conditions in attributing ethnic stereotypes to environmental causes was marginally significant, F(2, 1157) = 2.82, $p$ = .060, $\eta^2$ = .005. As in Study 1, participants in the Genetic Differences condition attributed the ethnic stereotypes marginally less to environmental causes ($M$ = 69.98%, $SD$ = 16.71%) than participants in the Genetic Similarities condition ($M$ = 72.56%, $SD$ = 17.86%), $p$ = .084, $d$ = 0.15. Participants in the Control condition did not differ in their tendencies to attribute ethnic stereotypes to environmental causes ($M$ = 72.19%, $SD$ = 15.26%) either from those in the Genetic Similarities condition, $p$ = .768, $d$ = 0.02, or from those in the Genetic Differences condition, $p$ = .137, $d$ = 0.14.

Analyzing the correlations among the measures revealed that the higher participants scored on genetic essentialist tendencies, the more they attributed the ethnic stereotypes to genetic causes both when using the Genetic Essentialist Tendencies scale, $r$ = .41, $p < .001$, and when using the Beliefs in Genetic Determinism scale, $r$ = .47, $p < .001$ (see Table 2; for these analyses,

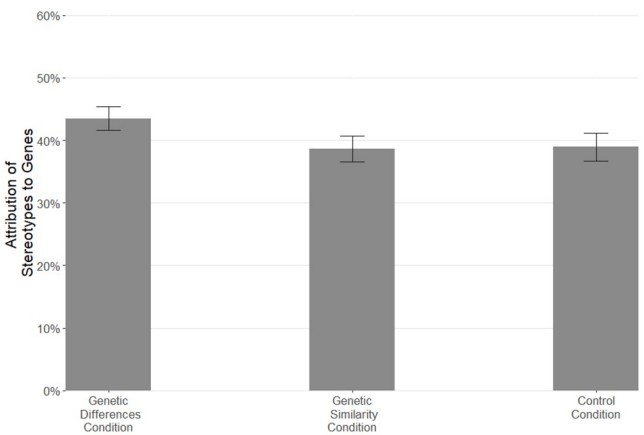

**Fig 2. Mean attribution of stereotypes to genes by condition.** Error bars are 95% confidence intervals.

all participants are included, regardless of whether they viewed the stereotypes as accurate). Likewise, genetic essentialist tendencies were also negatively correlated with attributions to environmental causes, $r = -.18$, $p < .001$ and $r = -.14$, $p < .001$, for Genetic Essentialist Tendencies and Beliefs in Genetic Determinism, respectively. In addition, the higher participants

**Table 2. Correlations between perceived accuracy of stereotypes, offensiveness of stereotypes, attribution of stereotypes to genes, attribution of stereotypes to environment, genetic essentialism tendencies, belief in genetic determinism, social dominance orientation, knowledge, and feelings towards ethnic groups.**

| | Perceived Accuracy of Stereotypes | Offensiveness of stereotypes | Attribution of stereotypes to genetic causes | Attribution of stereotypes to environmental causes | Genetic Essentialism Tendencies | Belief in Genetic Determinism | Social Dominance Orientation | Genetic Knowledge | Feelings towards the different ethnic groups |
|---|---|---|---|---|---|---|---|---|---|
| *Perceived Accuracy of Stereotypes* | | -0.19(< .001) | 0.47(< .001) | 0.05(.555) | 0.36(< .001) | 0.30(< .001) | 0.29(< .001) | -0.12(< .001) | -0.12(< .001) |
| *Offensiveness of stereotypes* | -0.19(< .001) | | -0.12(< .001) | 0.07(.098) | -0.05(.468) | -0.16(< .001) | -0.20(< .001) | 0.02(>.999) | 0.05(.584) |
| *Attribution of stereotypes to genetic causes* | 0.47(< .001) | -0.12(< .001) | | -0.29(< .001) | 0.41(< .001) | 0.47(< .001) | 0.25(< .001) | -0.29(< .001) | -0.10(.008) |
| *Attribution of stereotypes to environmental causes* | 0.05(.555) | 0.07(.098) | -0.29(< .001) | | -0.18(< .001) | -0.14(< .001) | -0.15(< .001) | 0.24(< .001) | 0.02(>.999) |
| *Genetic Essentialism Tendencies* | 0.36(< .001) | -0.05(.468) | 0.41(< .001) | -0.18(< .001) | | 0.57(< .001) | 0.25(< .001) | -0.23(< .001) | -0.08(.037) |
| *Belief in Genetic Determinism* | 0.30(< .001) | -0.16(< .001) | 0.47(< .001) | -0.14(< .001) | 0.57(< .001) | | 0.15(< .001) | -0.19(< .001) | -0.04(.940) |
| *Social Dominance Orientation* | 0.29(< .001) | -0.20(< .001) | 0.25(< .001) | -0.15(< .001) | 0.25(< .001) | 0.15(< .001) | | -0.12(< .001) | -0.29(< .001) |
| *Genetic Knowledge* | -0.12(< .001) | 0.02(>.999) | -0.29(< .001) | 0.24(< .001) | -0.23(< .001) | -0.19(< .001) | -0.12(< .001) | | 0.01(>.999) |
| *Feelings towards the different ethnic groups* | -0.12(< .001) | 0.05(.584) | -0.10(.008) | 0.02(>.999) | -0.08(.037) | -0.04(.940) | -0.29(< .001) | 0.01 (>.999) | |

Computed correlation used pearson-method with listwise-deletion.

scored on social dominance orientation, the more they attributed the ethnic stereotypes to genetic causes, $r = .25$, $p < .001$, and the less they did to environmental causes, $r = -.15$, $p < .001$. Interestingly, knowledge about genes was associated with less of a tendency to attribute stereotypes to genetic causes, $r = -.29$, $p < .001$, and a greater tendency to attribute them to environmental causes, $r = .24$, $p < .001$.

The data files, codebooks, and analysis scripts for both studies are available at https://osf.io/rdqw3/.

Finally, we wanted to explore whether participants in the genetic differences condition would show less positive attitudes towards people from the ethnic groups listed in the stereotypes. However, the overall ANOVA was not significant, $F(2, 1147) = 0.59$, $p = .557$, $\eta^2 = .001$, and there were no significant differences between participants in the Genetic Differences condition ($M = 69.48\%$, $SD = 17.37\%$), the Genetic Similarity condition ($M = 70.76\%$, $SD = 15.93\%$), or the Control condition ($M = 70.08\%$, $SD = 17.40\%$; all $ps > .05$).

The results largely replicated those from Study 1. People who reflected on the common genes that humanity shared, were less likely to interpret ethnic stereotypes in terms of underlying genetics compared to participants who reflected on genetic variation across the globe. One key difference between Studies 1 and 2 was that in Study 1 the results of the Control condition were more similar to those in the Genetic Differences condition, whereas in Study 2 those in the Control condition more closely resembled those in the Genetic Similarities condition. It is unclear why the Control condition varied across studies; perhaps because the stereotypes in Study 2 contained a larger number of negative stereotypes it is possible that people viewed more negative stereotypes to be more likely the product of environmental as opposed to genetic influences. In both studies, people's estimates for the environmental contribution to ethnic stereotypes was largely unaffected by our manipulations.

Study 2 also replicated our findings that people who score higher on measures of genetic essentialism are more likely to attribute stereotypes to genetic causes, and are less likely to attribute them to environmental causes. Similarly, across both studies people higher in social dominance orientation also attributed stereotypes more to genetic and less to environmental causes (although the correlation with environmental causes was not significant in Study 1). Furthermore, we found that people who had more knowledge about genes were less likely to attribute ethnic stereotypes to genetic causes, and were more likely to view them as the product of environmental causes. Finally, participants who perceived the stereotypes as more accurate, who attributed them more to genetic causes, who were higher on genetic essentialism (although this was only significant for one of the two measures), and who scored higher on social dominance orientation, had less positive feelings towards the different ethnic groups.

## General discussion

These results further the notion that people's understanding about genes can have broad implications for how they understand other aspects of their lives [4, 19]. Across two studies we found that people were more likely to view genes as underlying ethnic differences under the following conditions: a) when people had recently read an article describing how people's ancestry can be assessed by examining their genomes (in contrast to those who read an article describing the homogeneity of the human genome, and in contrast to those in a control group in Study 2); b) when people tend to have more deterministic and essentialist views of genes in general; and c) when people have relatively less general knowledge about genes. People's thoughts about genetics thus contribute to the ways they understand ethnic stereotypes. Moreover, the content in our Genetics Differences article was similar in kind to that conveyed by the advertisements of genetic ancestry companies, suggesting that the existence of this service

may be contributing to a more genetically based view of ethnic stereotypes. Such findings are important because other research has found that considering a biological basis of ethnic identity is often associated with a number of negative consequences [30, 35, 36], although we note that we did not find that our manipulations were associated with any differences in the warmth of attitudes towards other ethnic groups.

Previous work has found that encounters with genetic information can affect the ways that people think about ethnicity. For example, reading about genetic ancestry led Germans to have an ingroup preference for Western Europeans over Eastern Europeans [30], reading about variability in the human genome led participants to evaluate ingroup and outgroup faces in a more dichotomous way [31], and reading about genetic similarities between Jews and Arabs led American Jewish participants to support peacemaking efforts in the Middle East less [33]. While our study didn't find any difference in attitudes towards the different ethnic groups, it points to a possible reason for the findings from previous research: When people think about the genetic differences between different populations, they come to think of *any* differences between those populations as being caused more by genes. Because genes tend to be seen as ultimate causes, immutable, natural and as carving the social world up into homogeneous and discrete categories [19], different populations come to be seen as fundamentally different.

It is encouraging that genetics knowledge was associated with a tendency to view ethnic stereotypes to have less of a genetic foundation and more of an environmental one [also see 44]. This suggests that educating people about genetics may be an avenue for reducing people's more harmful stereotypes about other groups [45, 46].

## Limitations

This research relied upon presenting people with a list of ethnic stereotypes and asking them to consider the genetic and environmental causes to these. It is possible that there are demand characteristics in this method that resulted in people estimating a larger proportion of genetic influences than they would have spontaneously considered on their own. However, demand characteristics could not explain why people with different attitudes, knowledge, and those who read the Genetics Differences article would perceive a larger genetic foundation to these ethnic stereotypes.

Furthermore, the two experimental manipulations focused solely on the role of genes in the similarity/difference among different populations. While this design allowed us to distinguish the effects of perceiving different populations as genetically similar or different on people's attributions of stereotypes to genetic causes, it cannot speak to the role of perceived cultural similarities/differences between populations. It is possible that emphasizing cultural differences or similarities would also influence people's judgments of the causes of ethnic stereotypes.

In addition, while we tried to make the essays convincing and based on accurate scientific explanations, the two manipulations are not precise polar opposites of each other. For example, the genetic similarity condition describes explicitly that there is relatively little genetic variation among human populations while the genetic difference condition describes how ancestry can be traced based on genetic maps of the world that show how people from different parts of the world are genetically distinct. Hence, direct comparisons of the potency of the two experimental essays are complicated by the different kinds of information that each essay contains.

While we have demonstrated here that genetic essentialist beliefs are correlated with the ways that people conceive of population differences, we have not provided any evidence regarding where these beliefs come from. There are likely some cognitive and motivational precursors of genetic essentialist beliefs, and this remains an important question for future research.

These studies were conducted with American MTurk workers and it's not clear how well the results would generalize to samples from other cultural backgrounds, or with varying knowledge about genetics. Research conducted with other samples would be informative.

## Supporting information

**S1 File.**
(DOCX)

## Author Contributions

**Conceptualization:** Steven J. Heine.

**Data curation:** Anita Schmalor, Benjamin Y. Cheung.

**Formal analysis:** Anita Schmalor, Benjamin Y. Cheung.

**Funding acquisition:** Steven J. Heine.

**Resources:** Steven J. Heine.

**Supervision:** Steven J. Heine.

**Writing – original draft:** Anita Schmalor.

**Writing – review & editing:** Benjamin Y. Cheung, Steven J. Heine.

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
