## [Decision Letter · Decision Letter 0]

17 Nov 2020

PONE-D-20-30825

Exploring people’s thoughts about the causes of ethnic stereotypes

PLOS ONE

Dear Dr. Heine,

Thank you for submitting your manuscript to PLOS ONE. After careful consideration, we feel that it has merit but does not fully meet PLOS ONE’s publication criteria as it currently stands. Therefore, we invite you to submit a revised version of the manuscript that addresses the points raised during the review process.

We look forward to receiving your revised manuscript.

Kind regards,

Heming Wang, PhD

Academic Editor

PLOS ONE

Journal Requirements:

2. Thank you for including your ethics statement:  "University of British Columbia Behavioral Research Ethics Board H13-01354".   

Please amend your current ethics statement to confirm that your named institutional review board or ethics committee specifically approved this study.

"No."

Reviewers' comments:

Reviewer's Responses to Questions

**Comments to the Author**

1. Is the manuscript technically sound, and do the data support the conclusions?

Reviewer #1: Yes

Reviewer #2: Partly

2. Has the statistical analysis been performed appropriately and rigorously? 

Reviewer #1: Yes

Reviewer #2: Yes

3. Have the authors made all data underlying the findings in their manuscript fully available?

Reviewer #1: Yes

Reviewer #2: Yes

4. Is the manuscript presented in an intelligible fashion and written in standard English?

Reviewer #1: Yes

Reviewer #2: Yes

5. Review Comments to the Author

Reviewer #1: Review of "Exploring people’s thoughts about the causes of ethnic stereotypes"

The authors present two studies in which they manipulated (via reading materials) participants’ beliefs about human genetic homogeneity versus heterogeneity and then measured participants’ endorsement of various ethnic stereotypes and attributions of these stereotypes of genetic and experiential causes. In both studies, participants in the Genetic Difference condition (compared to those in the Genetic Similarity condition) displayed a stronger tendency to attribute ethnic stereotypes to genetic causes.

I appreciate the authors’ well-executed studies with high statistical power. I also appreciate the authors’ clear and thorough presentation of the relevant literature. I do, however, have a number of suggestions that I feel would improve the paper.

1. The authors provide a nice review of the existing literature on beliefs about biology/genetics and ethnic stereotyping. Indeed, a good deal of studies by researchers like Keller, Chao, Plaks, Eberhard, Halperin all point in the same direction: When people place more weight on biology/genetics, they are more likely to invoke ethnic stereotypes and more likely to endorse the accuracy of such stereotypes. For this reason, the authors could stand to articulate more explicitly how their studies are different from - and add something significant to - the existing literature.

2. I looked closely at the articles the authors used in the manipulation and noticed that there is no material about cultural variation. This is important because the authors seem keenly interested in pitting genetics against experience/culture in participants’ attributions. The fact that the manipulation is only on the genetic side of the equation means that this is not an ideal way to test which type of causation (genetic vs. experiential/cultural) looms larger in people’s minds. A more complete design would have included four articles that would have conformed to a 2x2 design: Genetic Similarity / Cultural Similarity, Genetic Similarity / Cultural Difference, Genetic Difference / Cultural Similarity, Genetic Difference / Cultural Difference. This would have allowed for a fairer assessment of participants’relative weighing of genetic vs cultural causes. In other words, if participants’ ratings were to show a greater difference (i.e., more persuasion) between the Difference vs. Similarity conditions on the Genetic dimension than on the Cultural Dimension, this would provide stronger evidence for the authors’ claims. But with a full 2 X 2 design, the data *could* turn out differently. It could reveal some interesting Gene X Culture interactions in laypeople’s minds on at least some of the stereotype items. For example, people might believe that Indians like spicy food for a combination of genetic *and* cultural reasons.

Indeed, many (most?) people likely hold some version of a simple hybrid model that involves combining both genetic and experiential/cultural components in an additive fashion. What differs between individuals is merely the percentages they assign to genes and culture.

Another thing I notice about the articles is that there are several, if not *confounds*, at least ways in which the two are not directly comparable. For example, although the Genetic Difference article discusses how human genetic variation could occur, it never actually mentions in an absolute sense how much of the entire genome varies from person to person or region to region. In contrast, the Genetic Similarity article *does* feature prominently material about how much overall genetic variation there is (i.e., very little). Thus, the manipulation is not really ‘little overall variation’ vs. ‘lots of overall variation’ - it’s more like ‘little variation’ vs. ‘silent’. Other differences between the articles: The Similarity article introduces two concepts (exponential growth and the relative recentness of homo sapiens as a distinct species) that are not mentioned at all in the Difference article. In other words, there is more being manipulated here than simply genetic sameness/difference. As such, we don’t know exactly what is causing the effect.

3. Regarding the data analyses, it would have been extremely helpful and informative if the authors had gone beyond reporting a few t-tests / correlation matrices. For example, they could have tested at least a few of the numerous potentially fascinating mediational models that are there in the data waiting to be tested. For example: genetic essentialism tendencies -> attribution to genetic causes -> stereotypes are accurate. Or: SDO->genetic essentialism tendencies -> attribution to genetic causes -> stereotypes are accurate. There are any number of additional mediational tests and moderated mediation tests that would provide additional insight into why the authors obtain their pattern they obtain.

In a related vein, I would have liked to see more discussion of the cognitive versus motivational origins of genetic essentialism. By ‘cognitive’ I mean the innocuous (but incorrect) beliefs that contribute to genetic essentialism. But ‘motivational’ I’m referring to the ideological/value systems that rely on genetic essentialism being true. One example of that is SDO. Another is probably Just World beliefs. In other words, well-meaning people with genuinely egalitarian aspirations may believe that people on different continents are genetically different simply because that’s the message they’ve received from their culture (media, education, family) and they’ve had no reason to dispute that. Thus, these people would be more likely to believe in some version of ‘separate but equal’. Others, however, may have a more ideologically-driven investment in believing that people from different parts of the world are genetically different. They may be more likely to believe in ‘separate and UNequal’.

This question of where the genetic beliefs come from has important implications for how easily people might be persuaded by articles like the ones the authors present. I suspect that people with a more ideologically-driven belief in either Genetic Difference or Genetic Similarity would be less persuadable by the opposing article than those who acquired their beliefs simply by passively absorbing messages from their culture.

Finally, I had a bit of an issue with how the authors used the word ‘stereotype’. I’ve just done a quick search and it turns out that Japan *is* at or near the top of the world in life expectancy and the Netherlands *is* at or near the top in average height. I raise this because the definition of the word ‘stereotype’ according to Webster’s dictionary is: “a widely held but fixed and oversimplified image or idea of a particular type of person or thing.” In other words, the ‘stereotype’ component only enters once you add in the genetic component - that’s what gives it the “fixed” and “oversimplified” aspect. But before you do that, (at least in the case of Japanese longevity, Dutch height, and several others on the list), it’s not so much a ‘stereotype’ as it is a correlation. Put differently, genetics takes a correlation (being Dutch - height) and turns it into causation (being Dutch *causes* height) - and THAT’S when it becomes a ‘stereotype’ in the true sense of the word.

On the whole, however, these are all fairly addressable concerns. I can see a future version of this paper being published in PLOSONE.

Reviewer #2: Summary and General Impression:

Across two studies, the researchers examined how ideas of genetic essentialism impact the perceived causes of ethnic stereotypes. In Study 1, participants read one of two articles which described the ability to estimate ancestry through genetics or the homogeneity of the human genome, or read a control article discussing an unrelated topic. They were then asked to describe the extent to which ethnic stereotypes were caused by genetic and environmental factors on a numerical scale. Among participants who believed ethnic stereotypes were accurate, those in the control condition and the ancestry via genetics condition viewed stereotypes as being more related to genetics and less related to environmental factors than those in the homogenous genome condition. Across conditions, however, participants rated environmental factors as the majority influence on ethnic stereotypes. Genetic essentialist beliefs were also associated with greater attribution of genetics to stereotypes.

In Study 2, the authors replicated Study 1 with some additions. They expanded their list of stereotypes, included a measure of knowledge about genetics, and measured attitudes towards the ethnic groups described in the list of stereotypes. The finding that those in the genetic ancestry condition were more likely to attribute stereotypes to genetic causes was replicated; however, those in the control condition were most similar to the homogenous genome condition, unlike Study 1.

Overall, the manuscript is well-written and well-organized. It provides an interesting extension to past research on genetic essentialism and racial attitudes by examining the influence of genetic information on perceptions of the origins of stereotypical behavior. Both studies use large sample sizes and the procedures for excluding participants are well-explained. The statistics used are appropriate and corrections for multiple comparisons and effect sizes are clearly present. The authors present interesting ideas on the real-world implications of this research given the popularity of ancestry-related genomic testing and the ability for increased knowledge about genes as a means of reducing essentialist attitudes.

Suggestions for Improvement

1. The general discussion discusses three major findings: that reading about genetic differences used to determine ancestry increases essentialist views of stereotypes, that people have essentialist views of genetics in general, and that overall genetic knowledge is low. In my read, the support for the first conclusion is strong. However, the scores for the first genetic essentialism scale are just above the midpoint in both studies. I would appreciate more explanation of how this finding is interpreted as generally more essentialist views.

2. Similarly, the authors state that overall genetic knowledge is low. This was measured only in Study 2, with a score of just over 6 / 9 correct. While this does not indicate a high level of knowledge about genes, I don’t know that I would characterize this as a low level of knowledge.

3. Overall, the discussion is quite short. I’d like to see some discussion of how the results of this research build upon and/or replicate previous work in this area.

4. Both studies include an analysis of the effects of social dominance orientation. However, it is not explained why this variable is included and social dominance orientation is not discussed in the introduction.

5. Information about participants’ racial/ethnic backgrounds is not provided. If this is available, it should be included in the participants section.

6. I would appreciate more information about the development of the stereotypes for both studies. How were these stereotypes selected? Was there any consideration of differential interpretations for stereotypes related to physical appearance compared to stereotypes about habits, personality, etc. It seems it would be easier to mentally link some of these stereotypes to genes compared to others.

7. Minor – There are numerous issues with APA format, particularly in the References section

6. PLOS authors have the option to publish the peer review history of their article (what does this mean?). If published, this will include your full peer review and any attached files.

Reviewer #1: No

Reviewer #2: No

---

## [Author Response · Author response to Decision Letter 0]

2 Dec 2020

Dear Dr. Wang,

Thank you for allowing us to send a revised version of our manuscript for consideration. We greatly appreciate the opportunity and we are also grateful to the reviewers for providing constructive comments that have helped us make this a better manuscript.

We outline, in point form, our responses to your and the reviewers’ comments along with any changes that we have made. We respond to them in the order that they appeared in your decision letter to us. 

Editor

Please provide additional details regarding participant consent. In the ethics statement in the Methods and online submission information, please ensure that you have specified (1) whether consent was informed and (2) what type you obtained (for instance, written or verbal, and if verbal, how it was documented and witnessed). If your study included minors, state whether you obtained consent from parents or guardians. If the need for consent was waived by the ethics committee, please include this information.

We added the following to the Methods Sections in both studies (pp. 10 & 16):

“Participants provided informed written consent. All participants were over 18.”

Thank you for stating the following in your Competing Interests section: 

"No."

Note that we have declared that no competing interests exist. 

Reviewer 1

1. The authors provide a nice review of the existing literature on beliefs about biology/genetics and ethnic stereotyping. Indeed, a good deal of studies by researchers like Keller, Chao, Plaks, Eberhard, Halperin all point in the same direction: When people place more weight on biology/genetics, they are more likely to invoke ethnic stereotypes and more likely to endorse the accuracy of such stereotypes. For this reason, the authors could stand to articulate more explicitly how their studies are different from - and add something significant to - the existing literature.

We thank the reviewer for this suggestion, and we have now added a section to the introduction where we discuss how our study is different from and adds to the existing literature (p. 8):

“While the extant literature has found that genetic information can affect the ways people think about different ethnicities, we explore whether thinking about genetic differences between different populations leads people to think of any differences between these populations as being more likely the result of genes. Because genes are often perceived as carving up the social world into homogenous and discrete categories (Dar-Nimrod & Heine, 2011), different populations may be seen as fundamentally different which might increase the psychological distance people feel towards them.”

2. I looked closely at the articles the authors used in the manipulation and noticed that there is no material about cultural variation. This is important because the authors seem keenly interested in pitting genetics against experience/culture in participants’ attributions. The fact that the manipulation is only on the genetic side of the equation means that this is not an ideal way to test which type of causation (genetic vs. experiential/cultural) looms larger in people’s minds. A more complete design would have included four articles that would have conformed to a 2x2 design: Genetic Similarity / Cultural Similarity, Genetic Similarity / Cultural Difference, Genetic Difference / Cultural Similarity, Genetic Difference / Cultural Difference. This would have allowed for a fairer assessment of participants’relative weighing of genetic vs cultural causes. In other words, if participants’ ratings were to show a greater difference (i.e., more persuasion) between the Difference vs. Similarity conditions on the Genetic dimension than on the Cultural Dimension, this would provide stronger evidence for the authors’ claims. But with a full 2 X 2 design, the data *could* turn out differently. It could reveal some interesting Gene X Culture interactions in laypeople’s minds on at least some of the stereotype items. For example, people might believe that Indians like spicy food for a combination of genetic *and* cultural reasons.

Indeed, many (most?) people likely hold some version of a simple hybrid model that involves combining both genetic and experiential/cultural components in an additive fashion. What differs between individuals is merely the percentages they assign to genes and culture.

Another thing I notice about the articles is that there are several, if not *confounds*, at least ways in which the two are not directly comparable. For example, although the Genetic Difference article discusses how human genetic variation could occur, it never actually mentions in an absolute sense how much of the entire genome varies from person to person or region to region. In contrast, the Genetic Similarity article *does* feature prominently material about how much overall genetic variation there is (i.e., very little). Thus, the manipulation is not really ‘little overall variation’ vs. ‘lots of overall variation’ - it’s more like ‘little variation’ vs. ‘silent’. Other differences between the articles: The Similarity article introduces two concepts (exponential growth and the relative recentness of homo sapiens as a distinct species) that are not mentioned at all in the Difference article. In other words, there is more being manipulated here than simply genetic sameness/difference. As such, we don’t know exactly what is causing the effect.

We thank the reviewer for this insightful point. We have added two paragraphs to the discussion where we discuss the limitations of the manipulations (p. 25): 

“Furthermore, the two experimental manipulations focused solely on the role of genes in the similarity/difference among different populations. While this design allowed us to distinguish the effects of perceiving different populations as genetically similar or different on people’s attributions of stereotypes to genetic causes, it cannot speak to the role of perceived cultural similarities/differences between populations. It is possible that emphasizing cultural differences or similarities would also influence people’s judgments of the causes of ethnic stereotypes. 

In addition, while we tried to make the essays convincing and based on accurate scientific explanations, the two manipulations are not precise polar opposites of each other. For example, the genetic similarity condition describes explicitly that there is relatively little genetic variation among human populations while the genetic difference condition describes how ancestry can be traced based on genetic maps of the world that show how people from different parts of the world are genetically distinct. Hence, direct comparisons of the potency of the two experimental essays are complicated by the different kinds of information that each essay contains.”

3. Regarding the data analyses, it would have been extremely helpful and informative if the authors had gone beyond reporting a few t-tests / correlation matrices. For example, they could have tested at least a few of the numerous potentially fascinating mediational models that are there in the data waiting to be tested. For example: genetic essentialism tendencies -> attribution to genetic causes -> stereotypes are accurate. Or: SDO->genetic essentialism tendencies -> attribution to genetic causes -> stereotypes are accurate. There are any number of additional mediational tests and moderated mediation tests that would provide additional insight into why the authors obtain their pattern they obtain.

In a related vein, I would have liked to see more discussion of the cognitive versus motivational origins of genetic essentialism. By ‘cognitive’ I mean the innocuous (but incorrect) beliefs that contribute to genetic essentialism. But ‘motivational’ I’m referring to the ideological/value systems that rely on genetic essentialism being true. One example of that is SDO. Another is probably Just World beliefs. In other words, well-meaning people with genuinely egalitarian aspirations may believe that people on different continents are genetically different simply because that’s the message they’ve received from their culture (media, education, family) and they’ve had no reason to dispute that. Thus, these people would be more likely to believe in some version of ‘separate but equal’. Others, however, may have a more ideologically-driven investment in believing that people from different parts of the world are genetically different. They may be more likely to believe in ‘separate and UNequal’.

This question of where the genetic beliefs come from has important implications for how easily people might be persuaded by articles like the ones the authors present. I suspect that people with a more ideologically-driven belief in either Genetic Difference or Genetic Similarity would be less persuadable by the opposing article than those who acquired their beliefs simply by passively absorbing messages from their culture.

We thank the reviewer for these thoughtful points. We have not done any of these mediational analyses as our paper is not intending to explore what predicts the perceived accuracy of stereotypes. While interesting, this seems to be a much different question from what we were investigating and would require a rather different experimental design to investigate it appropriately. We have now added the following brief paragraph to the discussion where we address the interesting point about the possible cognitive and motivational origins of genetic essentialism (pp. 26-27):

 “While we have demonstrated here that genetic essentialist beliefs are correlated with the ways that people conceive of population differences, we have not provided any evidence regarding where these beliefs come from. There are likely some cognitive and motivational precursors of genetic essentialist beliefs, and this remains an important question for future research.”

Finally, I had a bit of an issue with how the authors used the word ‘stereotype’. I’ve just done a quick search and it turns out that Japan *is* at or near the top of the world in life expectancy and the Netherlands *is* at or near the top in average height. I raise this because the definition of the word ‘stereotype’ according to Webster’s dictionary is: “a widely held but fixed and oversimplified image or idea of a particular type of person or thing.” In other words, the ‘stereotype’ component only enters once you add in the genetic component - that’s what gives it the “fixed” and “oversimplified” aspect. But before you do that, (at least in the case of Japanese longevity, Dutch height, and several others on the list), it’s not so much a ‘stereotype’ as it is a correlation. Put differently, genetics takes a correlation (being Dutch - height) and turns it into causation (being Dutch *causes* height) - and THAT’S when it becomes a ‘stereotype’ in the true sense of the word.

We feel we have used the word “stereotype” in ways that are consistent with the literature and with the Webster’s dictionary definition that the reviewer cites. There is a rather large literature that has explored the accuracy of different kinds of stereotypes (e.g., see the book “Stereotype accuracy: Towards appreciating group differences” by Lee, Jussim, & McCauley), which finds that many stereotypes do possess varying degrees of accuracy. In general, the field does not use different terms for those stereotypes which have an accurate basis and those which do not.

On the whole, however, these are all fairly addressable concerns. I can see a future version of this paper being published in PLOSONE.

Reviewer 2

1. The general discussion discusses three major findings: that reading about genetic differences used to determine ancestry increases essentialist views of stereotypes, that people have essentialist views of genetics in general, and that overall genetic knowledge is low. In my read, the support for the first conclusion is strong. However, the scores for the first genetic essentialism scale are just above the midpoint in both studies. I would appreciate more explanation of how this finding is interpreted as generally more essentialist views.

We agree with the reviewer that we cannot conclude from these studies that genetic essentialism in general is high and, indeed, we did not intend to draw that conclusion. Our intention was to highlight that our findings show that people view ethnic stereotypes to be more likely to be caused by genes three conditions: a) after reading an article about how people’s ancestry can be assessed by their genome, b) when people have more essentialist views, and c) when people have less knowledge about genes. We have now rephrased the discussion on these findings to make this conclusion clearer (p. 23):

“Across two studies we found that people were more likely to view genes as underlying ethnic differences under the following conditions: a) when people had recently read an article describing how people’s ancestry can be assessed by examining their genomes (in contrast to those who read an article describing the homogeneity of the human genome, and in contrast to those in a control group in Study 2); b) when people tend to have more deterministic and essentialist views of genes in general; and c) when people have relatively less general knowledge about genes.”

2. Similarly, the authors state that overall genetic knowledge is low. This was measured only in Study 2, with a score of just over 6 / 9 correct. While this does not indicate a high level of knowledge about genes, I don’t know that I would characterize this as a low level of knowledge.

We agree that our findings do not speak to the idea that genetic knowledge in general is low and we thank the reviewer for noting that our conclusion seemed to suggest this. Our intention here was not to say that genetic knowledge in general is low, but rather that when genetic knowledge is low, then people are more likely to view ethnic stereotypes as being caused by genes. We have now rephrased the discussion to make this clearer in point 1 above.

3. Overall, the discussion is quite short. I’d like to see some discussion of how the results of this research build upon and/or replicate previous work in this area.

We thank the reviewer for this suggestion, and we have now added a section to the discussion where we discuss how the results build on previous work (p. 24):

“Previous work has found that encounters with genetic information can affect the ways that people think about ethnicity. For example, reading about genetic ancestry led Germans to have an ingroup preference for Western Europeans over Eastern Europeans (Keller, 2005), reading about variability in the human genome led participants to evaluate ingroup and outgroup faces in a more dichotomous way (Plaks et al., 2012), and reading about genetic similarities between Jews and Arabs led American Jewish participants to support peacemaking efforts in the Middle East less. While our study didn’t find any difference in attitudes towards the different ethnic groups, it points to a possible reason for the findings from previous research: When people think about the genetic differences between different populations, they come to think of any differences between those populations as being caused more by genes. Because genes tend to be seen as ultimate causes, immutable, natural and as carving the social world up into homogeneous and discrete categories (Dar-Nimrod & Heine, 2011), different populations come to be seen as fundamentally different.” 

4. Both studies include an analysis of the effects of social dominance orientation. However, it is not explained why this variable is included and social dominance orientation is not discussed in the introduction. 

We thank the reviewer for calling our attention to this omission. We now describe this rationale on page 9.

“Last, because social dominance orientation is associated with more prejudicial attitudes towards outgroups (Pratto et al., 1994), we also explored how this variable would relate to people’s understandings about the causes of ethnic stereotypes.“

5. Information about participants’ racial/ethnic backgrounds is not provided. If this is available, it should be included in the participants section.

We thank the reviewer for calling our attention to this omission. We collected information about participants’ ethnic background for and added that to the:

Study 1 (pp. 9-10):

“We collected data from 425 Americans through MTurk in case some participants do not pass comprehension checks (discussed later; M age = 33.55, SD = 10.75; 52% male, 47% female, 1% other; 67% Caucasian, 6% Black/African American, 6% Hispanic, 5% Asian, 4% mixed, 3% other).”

Study 2 (p. 16):

“We collected data from 1238 Americans through MTurk in case some participants did not pass comprehension checks (M age = 34.53, SD = 11.16; 54% female, 45% male, 1% other; 69% Caucasian, 12% mixed, 8% Asian, 7% Black/African American, 2% Native American, 2% other).”

6. I would appreciate more information about the development of the stereotypes for both studies. How were these stereotypes selected? Was there any consideration of differential interpretations for stereotypes related to physical appearance compared to stereotypes about habits, personality, etc. It seems it would be easier to mentally link some of these stereotypes to genes compared to others.

We thank the reviewer for this oversight and have now added our rationale on p. 10. 

“We generated this list of stereotypes from internet searches, discussions with various individuals, and by reading a list of racial stereotypes (Chang & Kleiner, 2003).”

7. Minor – There are numerous issues with APA format, particularly in the References section

We followed the submission guidelines which state that the references should be in Vancouver style, not in APA style. 

We thank you for your very helpful review. 

Steven Heine

---

## [Decision Letter · Decision Letter 1]

2 Jan 2021

Exploring people’s thoughts about the causes of ethnic stereotypes

PONE-D-20-30825R1

Dear Dr. Heine,

We’re pleased to inform you that your manuscript has been judged scientifically suitable for publication and will be formally accepted for publication once it meets all outstanding technical requirements.

Kind regards,

Heming Wang, PhD

Academic Editor

PLOS ONE

Additional Editor Comments (optional):

Reviewers' comments:

Reviewer's Responses to Questions

**Comments to the Author**

1. If the authors have adequately addressed your comments raised in a previous round of review and you feel that this manuscript is now acceptable for publication, you may indicate that here to bypass the “Comments to the Author” section, enter your conflict of interest statement in the “Confidential to Editor” section, and submit your "Accept" recommendation.

Reviewer #1: All comments have been addressed

2. Is the manuscript technically sound, and do the data support the conclusions?

Reviewer #1: Yes

3. Has the statistical analysis been performed appropriately and rigorously? 

Reviewer #1: Yes

4. Have the authors made all data underlying the findings in their manuscript fully available?

Reviewer #1: Yes

5. Is the manuscript presented in an intelligible fashion and written in standard English?

Reviewer #1: Yes

6. Review Comments to the Author

Reviewer #1: I am satisfied with the authors' responses to my suggestions in the previous round of review. I believe this article would make a good addition to PLOS ONE.

7. PLOS authors have the option to publish the peer review history of their article (what does this mean?). If published, this will include your full peer review and any attached files.

Reviewer #1: No

---

## [Editor Report · Acceptance letter]

6 Jan 2021

PONE-D-20-30825R1 

Exploring people’s thoughts about the causes of ethnic stereotypes 

Dear Dr. Heine:

I'm pleased to inform you that your manuscript has been deemed suitable for publication in PLOS ONE. Congratulations! Your manuscript is now with our production department. 

Kind regards, 

on behalf of

Dr. Heming Wang 

Academic Editor

PLOS ONE